# Preparation and Anti-Mold Properties of Nano-ZnO/Poly(*N*-isopropylacrylamide) Composite Hydrogels

**DOI:** 10.3390/molecules25184135

**Published:** 2020-09-10

**Authors:** Jingjing Zhang, Qiuli Huang, Chungui Du, Rui Peng, Yating Hua, Qi Li, Ailian Hu, Junhui Zhou

**Affiliations:** School of Engineering, Zhejiang A&F University, Hangzhou 311300, China; jingjingzhang312@163.com (J.Z.); huangql@knt.cn (Q.H.); 18255276196@163.com (R.P.); artyhuahtl@163.com (Y.H.); LQ950011@163.com (Q.L.); hal15857832323@163.com (A.H.); zhoujunhui@oupaigroup.com (J.Z.)

**Keywords:** nano-ZnO, poly(*N*-isopropylacrylamide), temperature-sensitive hydrogel, bamboo, anti-mold properties

## Abstract

The aim of this study was to overcome drawbacks of the inhomogeneous dispersion and facile agglomeration of nano-ZnO/poly(*N*-isopropylacrylamide) composite hydrogels (nano-ZnO/PNIPAm composite hydrogels) during synthesis and improve the anti-mold property of the nano-ZnO/PNIPAm composite hydrogels. Here, nano-ZnO/PNIPAm composite hydrogels were prepared by the radical polymerization method. Fourier transform infrared (FTIR) spectroscopy, transmission electron microscopy (TEM), differential scanning calorimeter (DSC), and dynamic light scattering (DLS) were used to characterize the effects of different dispersants on the particle sizes, dispersions, and phase transition characteristics of the nano-ZnO/PNIPAm composite hydrogels. The anti-mold properties of nano-ZnO/PNIPAm composite hydrogels were studied. Results revealed that the nano-ZnO/PNIPAm composite hydrogel prepared by the addition of nano-ZnO dispersion liquid exhibited the smallest particle size, the most homogeneous dispersion, and the highest stability. The addition of the dispersant did not change the phase transition characteristics of nano-ZnO/PNIPAm, and the nano-ZnO/PNIPAm composite hydrogels (P_f_) exhibited good anti-mold properties to the bamboo mold.

## 1. Introduction

Poly(*N*-isopropylacrylamide) hydrogel (PNIPAm) is a typical temperature-sensitive smart hydrogel with a lower critical solution temperature (LCST) [1]. PNIPAm exhibits volume swelling or shrinkage in response to external temperature changes below its LCST [2]. Therefore, PNIPAm demonstrates wide application prospects for controlled drug release, enzyme reaction control, and bioengineering [3,4,5,6]. Recently, some researchers have performed a series of studies that exploit temperature-response characteristics of temperature-sensitive hydrogels to change the swelling-shrinking state for the controlled release of drugs [7,8,9,10,11,12,13]. In addition, there is an increasing trend in the study of PNIPAm antibacterial hydrogels [14,15]. In recent years, the research and application of inorganic antibacterial agents have attracted increasing attention. Among these agents, ZnO antibacterial agents exhibit non-toxicity, excellent antibacterial properties, non-mobility, and high thermal stability, etc. ZnO antibacterial agents have become a research hotspot of new antibacterial agents [16]. The combination of nano-ZnO and PNIPAm for the preparation of PNIPAm antibacterial hydrogels containing nano-ZnO is thought to demonstrate immense development and application prospects.

As a purely natural material, bamboo is extremely popular, and it is widely used in interior decoration, architecture, and other fields. However, bamboo forms mildew extremely easily, leading to the ~10% loss of the bamboo output [17]. Therefore, it is imperative to investigate the mildew prevention of bamboo. The most suitable temperature for the growth of bamboo mold is 26–32 °C [18]. As LCST occurs at a temperature of ~32 °C for PNIPAm [19], the nano-ZnO/PNIPAm composite hydrogel obtained by the capping of nano-ZnO in PNIPAm can be used for bamboo treatment. The temperature-sensitive characteristic is exploited to realize the controlled release of the bactericide nano-ZnO, making it possible to solve the mildew problem.

However, most of the current studies on nano-ZnO/PNIPAm composite hydrogels are mainly focused on optical and mechanical properties [20,21]. There have been few reports on the inhibition performance of bamboo molds. The author’s team carried out only the preliminary studies on the resistance of nano-ZnO/PNIPAm hydrogels to bamboo molds. Even more so, the study on nano-ZnO/PNIPAm hydrogels used to mildew proof of bamboo product has not been reported. Therefore, it is urgent to perform relevant systematic research. However, in previous studies, the author found that although both ZnO/PNIPAm hydrogel and ZnO/P(NIPAm-co-AAC) hydrogel had inhibitory effects on bamboo molds, the ZnO/PNIPAm hydrogel had weak inhibitory effects on bamboo molds [15]. This situation was probably because, due to its small size, high specific surface energy, and high specific surface area, nano-ZnO undergoes facile aggregation with the increase in particle size during its preparation, thereby reducing its antibacterial properties [22]. Thus, based on previous research, the effect of different dispersants on particle size and properties of the nano-ZnO/PNIPAm composite hydrogels were further investigated. Finally, the nano-ZnO/PNIPAm composite hydrogel prepared with the best properties was used to impregnate bamboo strips for mildew proof experiment.

## 2. Results and Analysis

In this study, four samples of nano-ZnO/PNIPAm composite hydrogels were synthesized using NIPAM monomer. The samples also included nano-ZnO dispersion (P_f_), nano-ZnO particles with the dispersant sodium dodecylbenzene sulfonate (SDBS) (P_s_), nano-ZnO particles with dispersant polyvinylpyrrolidone (PVP) (P_p_), and nano-ZnO particles with dispersant hexadecyltrimethylammonium bromide (CTAB) (P_c_). The results and analysis of this study are discussed below.

### 2.1. FTIR Analysis

Figure 1 shows the FTIR spectra of the *N*-isopropylacrylamide (NIPAm) and four samples of nano-ZnO/PNIPAm composite hydrogels.

Peaks located at 3505 cm^−1^ and 3300 cm^−1^ corresponded to the asymmetric stretching vibrations and symmetric stretching vibrations of N-H, respectively (Figure 1). The peak located near 3360 cm^−1^ corresponded to the stretching vibration of -OH. The peak located at 3077 cm^−1^ corresponded to the frequency-doubled stretching vibration of the strong amide band. Peaks located at 2974 cm^−1^, 2933 cm^−1^, and 2876 cm^−1^ corresponded to the methyl and methylene C-H vibrations. The peak observed at 1650 cm^−1^ corresponded to the characteristic peak for the amide I band’s C=O stretching vibration. The peak observed near 1548 cm^−1^ corresponded to the C-N stretching vibration of the amide II band. The peak observed near 1460 cm^−1^ corresponded to the -CH_3_ asymmetric bending vibration. The peaks located at 1386 cm^−1^ and 1368 cm^−1^ corresponded to the bis-methyl coupling split of -CH(CH_3_)_2_ symmetric vibration, while that located at ~1173 cm^−1^ corresponded to the contraction vibration peak of C-C in -CH(CH_3_)_2_. Vibration peaks located at ~980 cm^−1^ and 928 cm^−1^ were related to the end-vinyl groups. The results suggested that the composite hydrogel is mainly composed of a hydrophilic amide group (-CONH-) and a hydrophobic isopropyl group (-CH(CH_3_)_2_). In the FTIR spectrum of the monomer NIPAm, a -C=C- absorption peak was observed at 1662 cm^−1^; however, this peak disappeared in the FTIR spectra of P_f_, P_s_, and P_p_, indicative of successful polymerization. For the P_c_, a strong absorption peak was still observed at 1662 cm^−1^, suggesting that the free-radical polymerization of the NIPAm monomer does not occur. Therefore, Pc was not explored in subsequent experiments and was not mentioned in the results.

### 2.2. Dynamic Light Scattering (DLS) and Zeta Analysis

Figure 2 shows the particle size distributions and the changes in particle sizes with standing times. Table 1 shows the particle sizes and Zeta potential values of the ZnO/PNIPAm composite hydrogels after 3 h of static treatment.

The particle sizes of P_f_, P_s_, and P_p_ after ultrasonic dispersion were 130.5 nm, 246.5 nm, and 493.4 nm, respectively (Figure 2 and Table 1). P_f_ clearly exhibited the smallest particle size. With the increase in the standing time, the nano-ZnO/PNIPAm composite gels’ particle sizes gradually increased. A gentle upward trend was observed for P_f_ and Ps curves indicating that the increase in particle sizes of the two composite hydrogels reduces with the increase in the standing time, and it becomes more stable (Figure 2b). However, the P_p_ curve exhibited a relatively high slope with a rapid upward trend mainly because that the PVP added during the preparation was a non-ionic dispersant. It exhibited a weak adsorption effect on the positively charged nano-ZnO particles in an aqueous medium. The P_p_ particle size rapidly increased with time, eventually leading to an unstable P_p_ system that can undergo facile aggregation. The zeta potentials of P_f_, P_s_, and P_p_ were −15.94, −4.13, and 4.39, respectively (Table 1). The absolute zeta potential of P_f_ was the highest. The smaller sizes of the substances or dispersed particles, the higher the absolute zeta potentials, and the systems were more stable [23]. Hence, the P_f_ system is the most stable and most suitable for the application of the anti-mold agent.

### 2.3. TEM Analysis

Figure 3 shows TEM images highlighting the micro-morphologies of the nano-ZnO/PNIPAm composite hydrogels.

The nanoparticles of the three composite hydrogels of P_f_, P_s_, and P_p_ were clearly visible in the TEM images (Figure 3); however, P_p_ exhibited severe agglomeration. This agglomeration is related to the fact that the PVP added during the preparation of P_p_ is a non-ionic dispersant with a weak adsorption capacity for nano-ZnO, leading to the facile aggregation of P_p_. Compared with P_p_, less agglomeration and better dispersion of P_s_ and P_f_ were observed. Among the three hydrogels, P_f_ exhibited the least aggregation and the most uniform dispersion (Figure 3). This result is mainly related to the fact that SDBS added during the preparation of P_s_ is an anionic dispersant and negatively charged in water, while nano-ZnO particles are positively charged in an aqueous medium. Hence, SDBS and the hydrogel are attracted to each other by electrostatic interactions, affording a stable and dispersed gel system. The nano-ZnO dispersion added during the preparation of P_f_ easily combined with ZnO particles, affording a P_f_ system that is less likely to undergo agglomeration, with more uniform dispersion and higher stability.

### 2.4. DSC Analysis

Figure 4 shows DSC thermograms of nano-ZnO/PNIPAm composite hydrogels based on nano-ZnO dispersion (P_f_), nano-ZnO particles with dispersant PVP (P_p_) and with SDBS (P_s_), and DSC thermogram of nano-ZnO/PNIPAm hydrogel obtained from the previous experimental results [15].

The LCSTs of P_f_, P_s_, and P_p_ were ~32 °C, which was consistent with the LCST of PNIPAm at ~32 °C. Results indicated that the different types and amounts of dispersants added in this study do not change the phase transition character of ZnO/PNIPAm. This result is possibly related to the fact that, owing to the addition of SDBS, PVP, and nano-ZnO dispersion in the preparation of nano-ZnO/PNIPAm composite hydrogels, the ratio of hydrophilic groups to hydrophobic groups in the composite hydrogels systems did not change. The LCST of nano-ZnO/PNIPAm hydrogel was 32 °C [15]. Hence, the phase transition temperatures of the three composite hydrogels consistent with the previous experimental results of nano-ZnO/PNIPAm hydrogel were still 32 °C.

### 2.5. Anti-Mold Properties of Nano-ZnO/PNIPAm Composite Hydrogel Synthesized by the Addition of Nano-ZnO Dispersion(P_f_)

In this study, only anti-mold properties of nano-ZnO/PNIPAm composite hydrogel synthesized by the addition of nano-ZnO dispersion (P_f_) were discussed because P_f_ exhibits the highest stability, the smallest particle size, and the most homogeneous dispersion. Therefore, P_f_ was used to treat the bamboo strips to perform a mildew proofing experiment. Figure 5a,c show the infection levels of untreated and treated bamboo strips with P_f_. Figure 5b,d show the pictures of untreated and treated bamboo strips on the 28th day.

Bamboo strips in the control group (untreated with P_f_) were infected by *Penicillium citrinum* (PC), *Trichoderma viride* (TV), *Aspergillus niger* (AN), and a hybrid fungi group comprising PC, TV, and AN (Hun) in ~15 days, with an infection level 4 (Figure 5a). The bamboo strips treated with nano-ZnO/PNIPAm composite hydrogel synthesized by the addition of nano-ZnO dispersion (P_f_) were only infected by *Aspergillus niger* (AN) with an infection level of 1.67. In contrast, the infection levels with *Penicillium citrinum* (PC), *Trichoderma viride* (TV), and mixed mildew of the three mildews (Hun) were less than 1 (Figure 5c). Hence, P_f_ exhibits a relatively good mildew resistance. The photographs in Figure 5b,d clearly show the results. Therefore, after the treatment of bamboo strips with P_f_, their anti-mold properties significantly improved.

## 3. Materials and Methods

### 3.1. Materials

Analytically pure (AR) *N*-isopropylacrylamide (NIPAm) was commercially purchased from TCI Co., Ltd.; (Shanghai, China). AR-grade nano-ZnO particles, *N*,*N*′-methylenebisacrylamide (MBA), potassium persulfate (KPS), *N*,*N*,*N*′,*N*′-tetramethylethylenediamine (TEMED), sodium dodecylbenzene sulfonate (SDBS), hexadecyltrimethylammonium bromide (CTAB), polyvinylpyrrolidone (PVP, average Mw 360,000), and nano-ZnO dispersion (40 nm, 50 wt.% in H_2_O) were commercially purchased from Shanghai Aladdin Biochemical Technology Co.; Ltd (Shanghai, China).

Moso bamboo strips (length of 50 mm × width of 20 mm × thickness of 5 mm), with no knots and a moisture content of ~10%, was commercially obtained from Zhejiang Yongyu Bamboo Industry Co., Ltd. (Huzhou, China).

### 3.2. Method

#### 3.2.1. Preparation of Nano-ZnO/PNIPAm Composite Hydrogels

Different types of dispersants were used to prevent the particle sizes increase due to particle agglomeration. Nano-ZnO dispersion was added during the preparation of ZnO/PNIPAm composite hydrogels. Syntheses of the ZnO/PNIPAm composite hydrogels were as follows: first, 0.75 g of the NIPAM monomers were added into the 250-mL three-necked flask containing 100 mL of deionized water for complete dissolution. Secondly, different dispersants (0.05 g of SDBS, 0.1 g of CTAB, or 0.1 g of PVP) with 0.2 g of nano-ZnO particles, or only 300 μL of the nano-ZnO dispersion (containing 0.2 g of nano-ZnO particles) were added. (The usage of twice the amount of CTAB and PVP compared to SDBS because when 0.2 g of SDBS was added, the absolute value of Zeta potential is the largest, the particle size is the smallest, and the dispersion stability is the best [24]). Then, the cross-linking agent *N*,*N*′-methylene bis acrylamide (MBA) (5% of the monomer mass),the initiator potassium persulfate (KPS) (5% of the monomer mass), and the catalyst *N*,*N*,*N*′,*N*′-tetramethylethylenediamine (TEMED) (5% of the monomer mass) were sequentially added into the solution at 25 °C. The reactions were carried out for 6 h while stirring using a magnet. Finally, four samples of nano-ZnO/PNIPAm composite hydrogels were synthesized. They were named P_f_ (the nano-ZnO/PNIPAm composite hydrogels synthesized using NIPAM and nano-ZnO dispersion). P_s_ (nano-ZnO/PNIPAm composite hydrogels synthesized by NIPAM, nano-ZnO particles, and dispersant SDBS). P_p_ (nano-ZnO/PNIPAm composite hydrogels synthesized using NIPAM, nano-ZnO particles, and dispersant PVP). P_c_ (the nano-ZnO/PNIPAm composite hydrogels synthesized using NIPAM, nano-ZnO particles, and dispersant CTAB).

The synthesis processes are shown in Figure 6. The amounts of different dispersants, nano-ZnO particles, and nano-ZnO dispersion are shown in Table 2.

#### 3.2.2. Fourier Transform Infrared (FTIR) Spectroscopy Analysis

The freeze-dried nano-ZnO/PNIPAm composite hydrogels samples were grounded into a powder using a mortar. The powder was mixed with KBr in a mass ratio of 1:100 and pressed into a thin sheet. The molecular structures of the thin sheets were characterized and analyzed on a Nicolet 6700 FTIR spectrometer. The resolution and wavelength range for recording FTIR spectra were 4 cm^−1^ and 4000–400 cm^−1^.

#### 3.2.3. Dynamic Light Scattering (DLS) Analysis

The prepared nano-ZnO/PNIPAm composite hydrogels suspensions were diluted with deionized water after ultrasonic treatment for 3 min. Then, an appropriate amount of suspension was taken in a quartz sample cell. The surface of the sample cell was cleaned with a filter paper before being placed in a Zeta PALS-31484 instrument for measuring the particle size of the nano-ZnO/PNIPAm composite hydrogels. The measurement conditions were as follows: laser emission of 658 nm, laser energy of 4.0 mW, background scattering of 173°, and temperature of 25 °C. Every sample was scanned 5 times, and the average of the results was taken.

#### 3.2.4. Zeta Potential Analysis

First, appropriate amounts of the nano-ZnO/PNIPAm composite hydrogels solution were added into a quartz cuvette. The conductive plug was washed with deionized water and cleaned using a filter paper. Then, the plug was inserted into the cuvette, and the outer surface of the quartz cuvette was cleaned. Finally, the Zeta PALS-31484 instrument was utilized to examine the zeta potential of the composite hydrogels’ samples at a test temperature of 25 °C. Every composite hydrogel sample was repeated 5 times, and the average of the results was taken.

#### 3.2.5. Transmission Electron Microscopy (TEM) Analysis

First, a drop of the nano-ZnO/PNIPAm composite hydrogels suspensions was taken and placed on a copper mesh covered with a carbon film, followed by staining with 2 wt% uranyl acetate. After about 3 min, the excess solution was removed using a clean filter paper, followed by air-drying at room temperature. Finally, the microstructure of the nano-ZnO/PNIPAm composite hydrogels was observed and analyzed under an accelerating voltage of 80 kV by JEM-1200X TEM.

#### 3.2.6. Differential Scanning Calorimetry (DSC) Analysis

First, 8–10 mg of the nano-ZnO/PNIPAm composite hydrogels samples were weighed, and the phase transition temperatures of the nano-ZnO/PNIPAm composite hydrogels were analyzed on a Q2000 DSC system (TA Instruments, New Castle, DE, USA). The test temperature range was 20–50 °C, and deionized water was used as the reference.

#### 3.2.7. Anti-Mildew Test

First, bamboo strips were placed in a pressurized tank with a nano-ZnO/PNIPAm composite hydrogels solution synthesized by the addition of a nano-ZnO dispersion liquid, followed by immersion under a pressure of 0.3 MPa for 3 h and then removed and dried. Then, anti-mold properties of the nano-ZnO/PNIPAm composite hydrogels were examined according to the “Test method for anti-mildew agents in controlling wood mold and stain fungi” (GB/T 18261-2013) [25]: Spore suspensions of *Penicillium citrinum* (PC), *Trichoderma viride* (TV), *Aspergillus niger* (AN), and a hybrid fungi group comprising PC, TV, and AN (Hun) were smeared in Petri dishes containing the plate medium. After 2 min, every Petri dish put a sterilized U-shaped solid glass rod. Next, the Petri dishes were placed in an incubator at a temperature of 28 ± 2 °C and relative humidity of 85 ± 5% for mildew cultivation. After the molds were successfully cultivated, the bamboo strips were placed on the U-shaped glasses, and the edge of the Petri dishes was sealed with parafilm, repeated three times for each group. Finally, the Petri dishes with bamboo strips were placed in the incubator for a mildew resistance test. Every other day, the bamboo strips in the incubator infected by PC, TV, AN, and Hun were observed and recorded, and the infection values took the average of the results (see Table 3). On the 28th day, photographs of the bamboo samples were taken. The area of the bamboo strips infected by the fungus were observed and analyzed to determine the infection levels of the bamboo strips (Table 3). The anti-mold properties of the nano-ZnO/PNIPAm composite hydrogels were analyzed.

## 4. Conclusions

Nano-ZnO/PNIPAm composite hydrogels with different particle sizes and degrees of dispersion were prepared by the addition of different types of dispersants. In this study, four samples of nano-ZnO/PNIPAm composite hydrogels were synthesized using NIPAM monomer and by addition of: nano-ZnO dispersion (P_f_), nano-ZnO particles with dispersant sodium dodecylbenzene sulfonate (SDBS) (P_s_), nano-ZnO particles with dispersant polyvinylpyrrolidone (PVP) (P_p_) and nano-ZnO particles with dispersant hexadecyltrimethylammonium bromide (CTAB) (P_c_), nano-ZnO/PNIPAm composite hydrogel synthesized by the addition of nano-ZnO dispersion (P_f_) with the smallest particle size, the most homogeneous dispersion, and the highest stability was obtained by the addition of the nano-ZnO dispersion.

The phase transition characteristics of nano-ZnO/PNIPAm composite hydrogels were not changed by the addition of different types of dispersants in this experiment. The nano-ZnO/PNIPAm composite hydrogel obtained by the addition of nano-ZnO dispersion (P_f_) exhibited good anti-mold properties for *Aspergillus niger*, *Trichoderma viride*, *Penicillium citrinum*, and mixed mildew of the three mildews (Hun).

## Figures and Tables

**Figure 1 molecules-25-04135-f001:**
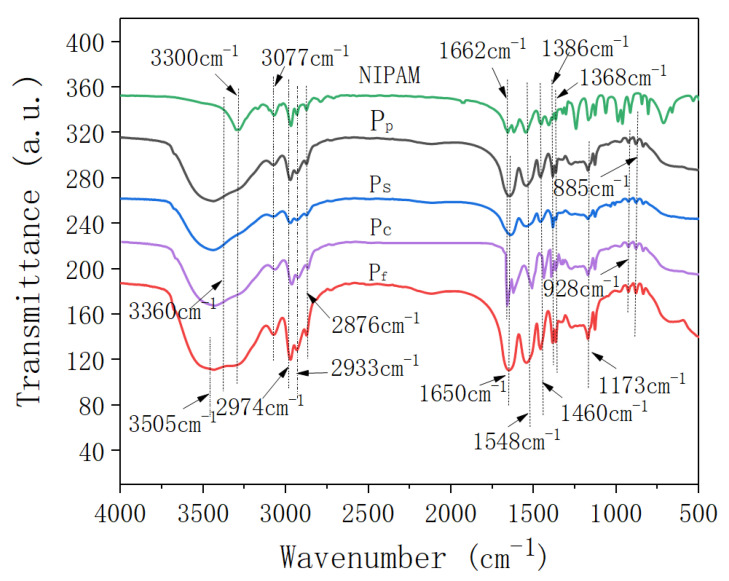
Infrared spectra of NIPAM monomer and the nano-ZnO/PNIPAm composite hydrogels.

**Figure 2 molecules-25-04135-f002:**
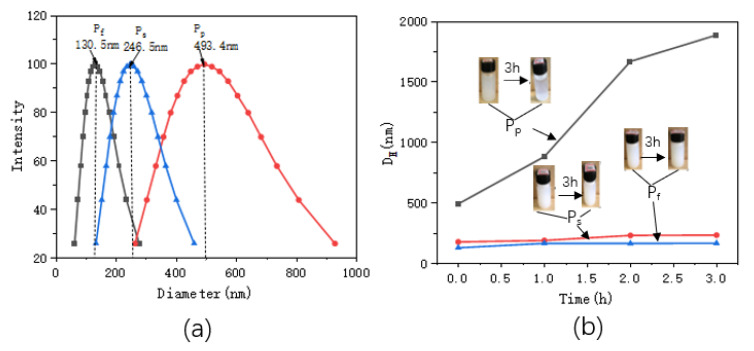
(**a**) distributions of the ZnO/PNIPAm composite hydrogels particle sizes, (**b**) relationships between the particle sizes and standing times.

**Figure 3 molecules-25-04135-f003:**
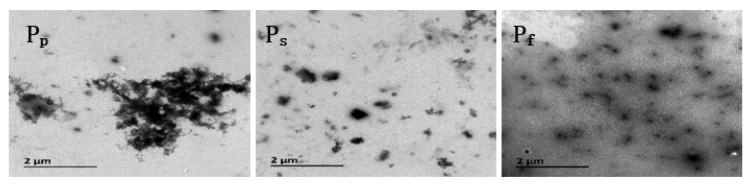
TEM images of nano-ZnO/PNIPAm composite hydrogels based on nano-ZnO particles with dispersant PVP (P_p_) nano-ZnO particles with dispersant SDBS (P_s_) and nano-ZnO dispersion (P_f_).

**Figure 4 molecules-25-04135-f004:**
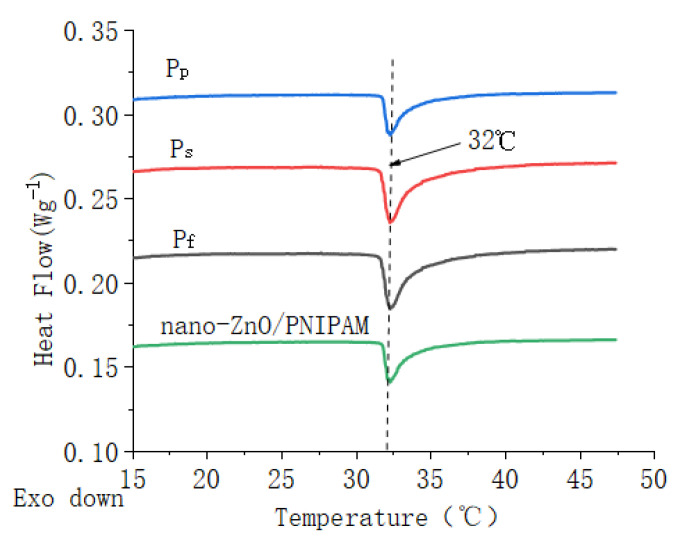
DSC curves of ZnO/PNIPAm composite hydrogels and nano-ZnO/PNIPAm hydrogel.

**Figure 5 molecules-25-04135-f005:**
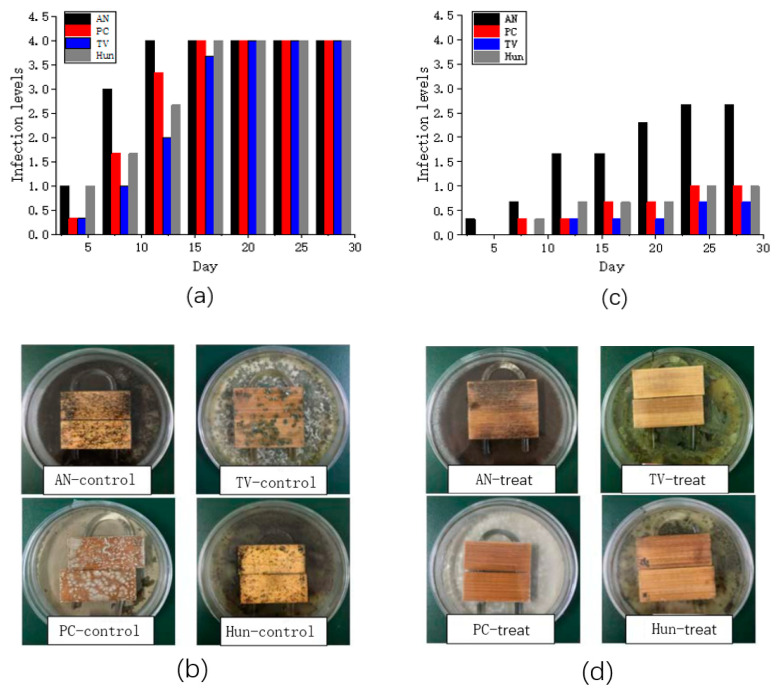
(**a**) the infection levels of the control group; (**b**) photographs of Petri dishes with the control group; (**c**) the infection levels of bamboo strips treated group with nano-ZnO/PNIPAm (Pf); (**d**) the photographs of Petri dishes of the treated group on the 28th day.

**Figure 6 molecules-25-04135-f006:**
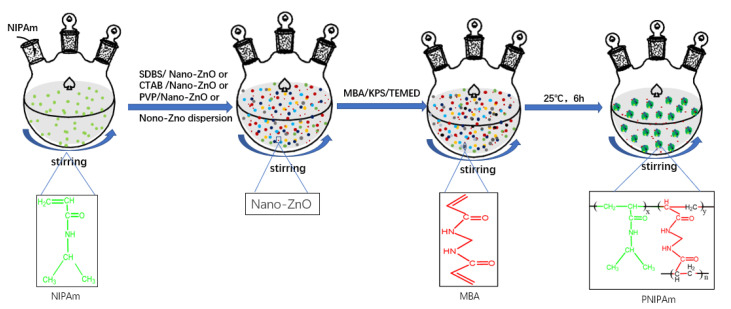
Schematic of the synthesis of the nano-ZnO/PNIPAm composite hydrogels.

**Table 1 molecules-25-04135-t001:** Average particle sizes and zeta potential values of ZnO/PNIPAm composite hydrogels.

Name of Composite Hydrogel	Initial Particle Size/nm	Particle Size after Standing for 3 h/nm	Zeta Potential/mV
P_f_	130.5	174.2	−15.94
P_s_	246.5	280.0	−4.13
P_p_	493.4	1595.4	4.39

**Table 2 molecules-25-04135-t002:** Amounts of nano-ZnO dispersants and nano-ZnO dispersion.

No.	Abbreviation of the Composite Hydrogel	Dispersant Name	Dispersant Mass/g	Nano-ZnO Particle Mass/g	Nano-ZnO Dispersion Volume/μL
1	P_f_	-	-	-	300
2	P_s_	SDBS	0.05	0.2	-
3	P_c_	CTAB	0.1	0.2	-
4	P_p_	PVP	0.1	0.2	-

Note: P_f_ (the nano-ZnO/PNIPAm composite hydrogels synthesized using NIPAM and nano-ZnO dispersion). P_s_ (nano-ZnO/PNIPAm composite hydrogels synthesized by NIPAM, nano-ZnO particles, and dispersant SDBS). P_p_ (nano-ZnO/PNIPAm composite hydrogels synthesized using NIPAM, nano-ZnO particles, and dispersant PVP). P_c_ (the nano-ZnO/PNIPAm composite hydrogels synthesized using NIPAM, nano-ZnO particles, and dispersant CTAB). There were 300 μL of the nano-ZnO dispersion containing 0.2 g of nano-ZnO particles.

**Table 3 molecules-25-04135-t003:** Classification standard of surface infection levels of samples.

Infection Levels	Infected Area of Sample
0	No hyphae or mildew on the sample surface
1	Infected area of sample < 1/4
2	Infected area of sample 1/4–1/2
3	Infected area of sample 1/2–3/4
4	Infected area of sample > 3/4

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
