# Peer review of "Preparation and Anti-Mold Properties of Nano-ZnO/Poly(N-isopropylacrylamide) Composite Hydrogels"

_molecules, 2020, doi:10.3390/molecules25184135_

Round 1
Reviewer 1 Report
The authors revised the manuscript well, and I would like to ask to provide the pH in which the zeta potential was measured.
Author Response
The authors revised the manuscript well, and I would like to ask to provide the pH in which the zeta potential was measured.
Response: Thanks for your kind question. The pH is 7.0.
Reviewer 2 Report
- It is needed to improve first sentence in the Abstract, as the sentence is not finished.
- "N,N'-methylenebisacrylamide" is one word.
- Please, provide average molar mass for used PVP.
- Lines 64-66 need to be deleted. They are only as instruction for authors.
- Line 84: Please, it is needed to rewrite sentence with addition of used dispersant types (SDBS CTAB and PVP).
- Lines 86-87: Please, it is needed to provide function in synthesis process for potassium persulfate and N,N,N',N'-tetramethylethylenediamine, because they are not cross-linking agents like as N,N'-methylenebisacrylamide.
- It is necessary for the authors to adopt simple abbreviations of composite hydrogel samples at the beginning in subsection 2.2.1. Preparation composite hydrogels and to apply it uniformly in the text.
- In Figure 1 is not clearly presented that only one dispersant (SDBS, CTA and PVP) was used as options for synthesis with nano-ZnO particles. It is needed improve this presentation in figure 1.
- In Table 1 is needed to insert new column with data of added nano-ZnO amount in the samples, as source of ZnO. Table 1. Amounts of nano-ZnO dispersants, nanoZnO particles and nano-ZnO dispersion.
- Line 94-98: Noted data, with abbreviated sample names, is needed to be explained also in the manuscript text in subsection 2.2.1. Preparation composite hydrogels, not only after Table 1. It is needed to improve sentence with data and full names of analyzed series of 4 composite hydrogels (Pf, Ps, Pc and Pp) in Figure 1.
- Why did the authors use twice higher amount of CTAB and PVP compared to SDBS?
- In Figure 2 following are missed:
- ordinate label name (Transmittance),
- FTIR spectrum of Pc sample in the whole range, as well as other spectra,
- in the caption of Fig. 2, FTIR spectrum of monomer NIPAM is missing,
and the line for peak located near 3360 cm−1 is slightly moved.
Please, correct all of them.
- Lines 156-158: The explanation "... (Ps) synthesized by the addition of dispersant sodium dodecylbenzene sulfonate (SDBS), (Pp) synthesized by the addition of the dispersant polyvinylpyrrolidone (PVP), and (Pc) synthesized by the addition of the dispersant hexadecyltrimethylammonium bromide (CTAB)". would be better to provide in part 3. Results and analysis, before 3.1. part, e.g.:
"Four samples of nano-ZnO/PNIPAm composite hydrogels were synthesized using NIPAM monomer and by addition of: nano-ZnO dispersion (Pf), nano-ZnO particles with dispersant sodium dodecylbenzene sulfonate (SDBS) (Ps), nano-ZnO particles with dispersant polyvinylpyrrolidone (PVP) (Pp) and nano-ZnO particles with dispersant hexadecyltrimethylammonium bromide (CTAB) (Pc)."
- Figure 3(b) is needed to improve, as the tags for Pp, Ps and Pf are duplicated!
- In Figure 4. It is needed to provide sample names for Pp, Ps and Pf, e.g. as a, b) and c) and if possible better quality of TEM.
- In sentence in line 227 and in Figure 5 caption it is needed to provide better explanation of samples names, e.g. DSC thermograms of ZnO/PNIPAm composite hydrogels based on: nano-ZnO dispersion (Pf), nano-ZnO particles with dispersant PVP (Pp) and with SDBS (Ps). Also, it is needed to provide if Exo is up or down in Heat Flow. It will be useful to present thermal characterization in region above 50°C with glass transitions.
- Lines 236-237: Authors have to provide data about reference with which compared previous experimental results of nano-ZnO/PNIPAm.
- Lines 238-256: It is needed to correct subsection name 3.5. Anti-mold properties of Pf nano-ZnO/PNIPAm composite hydrogel, according new abbreviated name.
- Firstly, it is needed to provide explanation why authors chose only the nano-ZnO/PNIPAm composite hydrogel sample prepared by addition of nano-ZnO dispersion, Pf, for further analysis, e.g. "Nano-ZnO/PNIPAm composite hydrogel sample prepared by addition of nano-ZnO dispersion, Pf, exhibits the highest stability, the smallest particle size and the most homogeneous dispersion..."
- Figure 6(a) and Figure 6(b) show the infection levels of untreated and treated bamboo strips with a nano-ZnO/PNIPAm composite hydrogels solution synthesized by the addition of a nano-ZnO dispersion, Pf.
- Line 251: It is needed to correct part of sentence: "The bamboo strips treated with the Pf"... with better explanation as: "The bamboo strips treated with nano-ZnO/PNIPAm composite hydrogel synthesized by the addition of nano-ZnO dispersion, Pf ".
- Lines 246-248: It is needed to improve organization and caption of Figure 6 with in-text explanation with addition of full data on infection levels after treatment of control and treated group of fungi (with full Latin names: Aspergillus niger (AN), Trichoderma viride (TV), Penicillium citrinum (PC), and mixed mildew of the three mildews (Hun). Also, it would be better to divide figure 6c in two parts, e.g.: Figure 6a) the infection levels of control group, 6b) photographs of Petri dishes with the control group, 6c) the infection levels of bamboo strips treated group with a nano-ZnO/PNIPAm, Pf (6d) the photographs of Petri dishes with treated group during 28 days.
- Line 260: Please, provide explanation for all four types prepared (Pf, Ps, Pc and Pp) composite hydrogels and then concluded which type was performed with the best properties.
- Line 264-266: It will be better to improve sentence as: "The nano-ZnO/PNIPAm composite hydrogel obtained by the addition of a nano-ZnO dispersion, Pf, exhibited good anti-mold properties for Aspergillus niger, Trichoderma viride, Penicillium citrinum, and mixed mildew of the three mildews (Hun)."
- Please, provide full name of the acronyms on first appearance in the main manuscript text and figures captions (Pf, Ps, Pc and Pp, PC, TV, AN and Hun).
- In the whole manuscript text there are many typographical and grammar mistakes that can be corrected, e.g. merged words.
- It is needed to check and correct all references according Journal`s Instruction with abbreviated journal names.
Reviewer 3 Report
The authors investigate the effect of different types of dispersants on the particle size, dispersibility and phase transition characteristics of nano-ZnO/PNIPAm composite hydrogels as well as the anti-mold properties of these hydrogels to the bamboo mold. The manuscript has been significantly improved compared to the previous version but still has some imperfections:
- The Figure 3b should be checked and corrected (double marks).
- For easy comparison, the maximum level of the infection scale in Fig. 6a and 6b should have the same value.
- There are a lot of missing spaces between words in the text. The manuscript should be carefuly checked and edited.
Round 2
Reviewer 2 Report
Manuscript molecules-908465 entitled “Preparation and anti-mold properties of nano-ZnO/poly(N-isopropylacrylamide) composite hydrogels” provide many improvements in new version manuscript text, but, it is needed to implement additional corrections before publishing.
- It is needed to provide details about which kind of average molar mass of PVP is provided (number average molar mass, Mn; mass average molar mass, Mw; Z average molar mass, Mz; or viscosity average molar mass, Mv), as well as unit (g/mol).
- I would recommended new captions, e.g.: "Figure 3. TEM images of nano-ZnO/PNIPAm composite hydrogels based on: nano-ZnO particles with dispersant PVP (Pp) nano-ZnO particles with dispersant SDBS (Ps) and nano-ZnO dispersion (Pf)".
- Lines 135-137: Authors should indicate that in Figure 4 they compared DSC curves of three nano-ZnO/PNIPAm composite hydrogels with the nano-ZnO/PNIPAm hydrogel from their previous work. Also, lines 146-147: Authors need to provide discussion of DSC results with previously published data (Huang, Q.L., Du, C.G., Hua, Y.T., Zhang, J., Peng,R., Yao, X.L. Synthesis and Characterization of Loaded Nano/Zinc Oxide Composite Hydrogels Intended for Anti-Mold Coatings on Bamboo. Bioresources.2019,14,7134-7147) and add it to the Reference list.
- In Figure 4 it is needed to insert "Exo down" for Heat Flow.
- Lines 187-189 need to be deleted. These are only instruction for authors.
- Line 191: "N- isopropylacrylamide ( NIPAm )" - Please, correct in manuscript text as N-isopropylacrylamide (NIPAm)
- Lines 206-208: In this sentence, procedure for obtaining all 4 types of composite hydrogels is still not clear. Please, it is needed to improve the procedure description for each sample, e.g. "Secondly, different dispersants (0.05 g of SDBS, or 0.1g of CTAB, or 0.1g of PVP) with 0.2 g of nano-ZnO particles, or only 300 μL of the nano-ZnO dispersion (containing 0.2g of nano-ZnO particles) were added.
- In inserted new column in Table 2, it is needed to correct data of added nano-ZnO amount per samples, as source of ZnO ("Nano-ZnO amount/g" not only "Nano-ZnO particle mass/g"), because sample 1 had also 0.2 g in 300 μL as stated in text. Please check data of its density, if mass of nano-ZnO dispersion is 0,2 g into 300 μL = 0,3 mL in Nano-ZnO particle.
- Lines 213-217 and 224-227: it would be more precise "... Pf (the nano-ZnO/PNIPAm composite hydrogels synthesized using NIPAM and nano-ZnO dispersion), Ps (nano-ZnO/PNIPAm composite hydrogels synthesized by NIPAM, nano-ZnO particles and dispersant SDBS), Pp (nano-ZnO/PNIPAm composite hydrogels synthesized using NIPAM, nano-ZnO particles and dispersant PVP) and Pc (the nano-ZnO/PNIPAm composite hydrogels synthesized using NIPAM, nano-ZnO particles and dispersant CTAB)."
- It would be useful to add in the manuscript, text explanation about usage of twice higher amount of CTAB and PVP compared to SDBS, according to mentioned paper (Pang, J. S., Zhang, H. Y., Cao, B., Song, C. W., Mao, L. B., Chen, J. (2009). Research on Dispersion of ZnO Nanoparticles in Aqueous Coating System. Bulletin of Chinese Ceramic Society, 01108-01112) and add it to the Reference list.
- In the whole manuscript text there are many typographical and grammar mistakes that can be corrected, e.g. merged words.
Author Response
Please see the attachment.

This manuscript is a resubmission of an earlier submission. The following is a list of the peer review reports and author responses from that submission.
Round 1
Reviewer 1 Report
The novelty of the article is very low. A lot of experiments are presented in other articles describing different acrylamide hydrogels doped with ZnO nanoparticles with anti mold properties on Bamboo plant (e.g."Synthesis and Characterization of Loaded Nano/Zinc Oxide Composite Hydrogels Intended for Anti-Mold Coatings on Bamboo", Qiuli Huang, Chungui Du, Yating Hua, Jingjing Zhang, Rui Peng, Xiaoling Yao, BioResources, vol. 14, No 3, 2019).
In the synthesis procedure there is no determination of polymerization rate (e.g. determination of unreacted monomer), so different hydrogels with dispersants cannot be compared appropriately.
The DLS and TEM analyses prove that the product is a micro dispersion, not the nano one (the particle size is above 100 nm). Moreover, on TEM images the magnification is too low and no ZnO particles can be observed, so there is no prove that there was no agglomeration of nanoparticles.
Authors should not describe ZnO nanoparticles as antibacterial in testing the antifungal properties, due to the fact that mold is a fungus not a bacteria.
Reviewer 2 Report
- The section order should be changed:
Section 3. Materials and methods should be immediately after section 1. Introduction and before section Results and analysis. - In Materials and method, there is described that authors used also dispersant CTAB. In the Table 2, there is mentioned that authors prepared four different samples, Pf,Ps, Pc and Pp. However, results of their findings with CTAB were not mentioned in whole paper. Why?
- During preparation of sample No.2, authors used less amount of dispersant SDBS. I would like to ask authors why did they lower the amount of SDBS (0.05g) when in the case of other dispersants (CTAB and PVP) they used two-times more (0.1g)?
- According to point 3, presented comparison in the article seems to be faulty for the reason that the different amount of used dispersant can affect the final behavior nanoparticles in creating nano-ZnO/PNIPAm hydrogel. This assumption is based on already published paper with ZnS and CTAB, when authors used different amount of surfactant CTAB to study stability of ZnS nanoparticles (source: DOI 10.1016/j.jcis.2012.03.073)
- Author used nano-ZnO (powder, 0.2g) in the case of samples No.2,3 and 4 and in the case of sample No.1 they used nano-ZnO dispersion (300 μL). What was concentration (mass of ZnO) in this 300μL of nano-ZnO dispersion? Was the powdered nano-ZnO obtained by vacuum drying, freeze-drying or thermal dyring of purchased nano-ZnO dispersion?
- In the section Materials, nano-ZnO was purchased already in the form of dispersion. Was nano-ZnO dispersed in water, ethanol or in something else? What was the concentration of nano-ZnO according to supplier?
- Line 175, authors write about preparation of sample No.1: "instead of dispersants, 300 μL of the nano-ZnO dispersion and nano-ZnO particles were added. In the Table 2, there is missing amount of nano-ZnO particles. Was it 0.2 g nano-ZnO as well?
- Figure 5 a),b) and c). Positioning of all images should be changed. It is confusing. I would suggest to authors: put Figure 5 a) and b) next to each other and Figure 5 c) immediately below a) and b). Write the description for the Figure 5 a), b) and c) in one paragraph below images.
- TEM images are in very poor quality
- Line 83 check "changes ,and", line 96 check "trend .This" line 98 check: "medium,so", line 111 "nonionic" to "non-ionic", line 116 "positivelycharged", line 221 "medium ,and" and "rodin", line 243 "experiments . All"
Based on the above mentioned problems of the presented paper I cannot recommend the submitted manuscript for publication in so good scientific journal as is Molecules.
Reviewer 3 Report
The paper presents interesting study on problem of inhomogeneous dispersion and facile agglomeration of nano-ZnO in PNIPAm hydrogel during synthesis. The authors investigate the effect of different types of dispersants on the particle size, dispersibility and phase transition characteristics of nano-ZnO/PNIPAm composite hydrogels as well as the anti-mold properties of these hydrogels to the bamboo mold. The manuscript presents interesting results, but requires minor revision.
Please find my comments:
- The locations of the Materials and Methods in section 3 (after Results) make the manuscript difficult to read because the names of the hydrogels and abbreviations of the reagents appear in the previous section without explanation. Abbreviations and names should be explained on first use. The order of sections should be rethought.
- The type of instruments of DLS (line 191), zeta potential (line 199) and TEM (line 207) are missing and should be added.
- Classification of infection levels presented in Table 4 covers only 4 levels while Figure 5 shows a more precise value for the level of infection and the text describes the level of infection to two decimal places. The signs for the degree of infection should be specified more precisely.
- What do the abbreviations Nano-ZnO and FS/nano-ZnO in Figure 6 mean?
- Was the anti-mildew test run in the dark? Can the anti-mold effect be a result of the photocatalytic properties of ZnO? It’s a pity the authors do not try to discuss the mechanism of the anti-mold action.
- Authors should compare the amounts of ZnO introduced as dispersion (in Pf) and particles (other composites). Without comparing the amount of ZnO in composite hydrogels, it is not known whether the difference in properties is due only to better dispersion or also to a greater amount of ZnO in the hydrogel.
- The effect of the amount of dispersants on the properties of the composite has not been investigated (only one amount was used for individual additives), so the conclusions should be rebuilt.
Reviewer 4 Report
Dear Editor,
Manuscript molecules-877192 entitled “Preparation and anti-mold properties of nano-ZnO/poly(N-isopropylacrylamide) composite hydrogels” written by Jingjing Zhang, Qiuli Huang, Chungui Du, Rui Peng, Yating Hua, Qi Li, Ailian Hu and Junhui Zhou, describes useful investigation about application of nano-ZnO/poly(N-isopropylacrylamide) composite hydrogels (nano-ZnO/PNIPAm) with improved anti-mold properties in the treated bamboo slices. A series of four nano-ZnO/PNIPAm composite hydrogels were prepared and the effects of different amounts and types of dispersants on the particle size, dispersibility, and phase transition characteristics were investigated. Nano-ZnO/PNIPAm composite hydrogel sample prepared by addition of nano-ZnO dispersion liquid exhibits the highest stability, the smallest particle size and the most homogeneous dispersion. This manuscript is part of authors´ wider investigation, it provided applicable results and may be published in the journal "Molecules" after following improvements:
- It is needed to improve Abstract with briefly description of the main applied methods and treatments.
- Lines 57-58: It is needed to improve sentence with data and full names of analyzed series of 4 composite hydrogels (Pf, Ps, Pc and Pp) in Figure 1. For better understanding and conclusion that Pc composite hydrogel was not polymerized, it will be useful to insert FTIR spectra of NIPAm monomer and Pc sample. Maybe it's just the remaining part of unreacted monomer?
- Line 60: It is needed to correct Figure 1 caption with addition of explanation with full names of abbreviations of composite hydrogels Pf, Ps, Pc and Pp.
- Lines 83-84, 86-89: Please, rewrite sentence for better definition which results were presented in figure 2 and check that there are a series of 3 hydrogels (Pf, Ps and Pp).
- Line 102, pay attention about in-text reference.
- Lines 122-123: Please, rewrite sentence for better definition which results were presented in figure 4 and check that there are a series of 3 hydrogels (Pf, Ps and Pp) - it is plural, not singular.
- In Figure 4. ordinate label is missing.
- Authors can provide discussion of DSC results with published data (e.g. in ref. 15). Why authors recorded DSC data only up to 50 °C? If is possible, it will be useful to present thermal characterization in wider region and maybe some differences in other transitions, e.g. glass transitions.
- Lines 126-133: Please, synchronize the names of "nano-ZnO/PNIPAm composite gels" and composite hydrogel, according to well-known definitions and differences between "gel" and "hydrogel".
- Line 133 - 5. Anti-mold properties of Pf nano-ZnO/PNIPAm composite hydrogel
- Lines 134-145: It is needed to better organize the items presented in Figure 5 (according Journal`s Instruction and Template) and for its in-text explanation with addition of full data on infection levels after treatment of control and treated group of fungi (with full italicized Latin names of Aspergillus niger (AN), Trichoderma viride (TV), Penicillium citrinum (PC), and mixed mildew of the three mildews (Hun).
"Figure 5. This is a figure, Schemes follow the same formatting. If there are multiple panels, they should be listed as: (a) Description of what is contained in the first panel; (b) Description of what is contained in the second panel. Figures should be placed in the main text near to the first time they are cited. A caption on a single line should be centered."
- Line 166: Please rewrite in plural: "3.2.1. Preparation of nano-ZnO/PNIPAm composite hydrogels".
- Lines 167-178: Please, provide clarification procedure for preparation of series of four composite hydrogels Pf, Ps, Pc and Pp. In my opinion, text in lines 172-178 will be at first, explanation of procedure to addition of dispersants need to follow, and then Table 2 should be placed at the end.
- Line 89 and 171: Please note about Journal`s Instruction and Template for Table 1 and 2
- Line 173: In the acronym "PNIPAm", letter "P" is usual abbreviation for polymer, not for monomer! Authors needed to clarify what was used - PNIPAm polymer or NIPAm monomer!
- Figure 6: It is needed to correct an error in NIPAm monomer structural formula. Figure 6 needed to be explained in manuscript text and to give full names of abbreviated terms: FS/nano-ZnO and Fnano-ZnO!
- Line 177: What is TMEDA? Is it probably typo error instead of TEMED?
- What is role of KPS? How polymerization process was initiated?
- Line 231: It is needed to capitalize word: Conclusions.
- Lines 233-240: Instead part of sentence: "Among the composite hydrogels,"... firstly, it will be better to add new sentence with notification that all four types of composite hydrogels were prepared and then concluded which type was performed with the best properties.
- The second sentence will be: "Nano-ZnO/PNIPAm composite hydrogel with the smallest particle size, the most homogeneous dispersion, and the highest stability was obtained by the addition of the nano-ZnO dispersion".
- "The phase transition characteristics of Nano-ZnO/PNIPAm composite hydrogels were not changed by addition of different types as well as different amounts of dispersants in this experiment. The Nano-ZnO/PNIPAm composite hydrogel exhibited good anti-mold properties for Aspergillus niger, Trichoderma viride, Penicillium citrinum, and mixed mildew of the three mildews (Hun)."
- In the whole manuscript text there are many typographical and grammar mistakes that can be corrected (e.g. merged words, singular instead plural, ...lines: 9, 11, 24, 33, 83-84, 96,102, 182, 188, 200, 209, 231).
- Please, provide full name of the acronyms on first appearance in the main manuscript text and figures 1,2,3,4, captions (e.g. Pf, Ps, Pc and Pp).
- Please, rewrite Latin names of applied fungi italicized in the manuscript text.
- It will be useful to improve discussion of all obtained results and compare with previously published literature data, especially with 15.
- Please, provide doi numbers when possible. Some of references are impossible to find. It is needed to check and correct all references according Journal`s Instruction and Template:
1. Author 1, A.B.; Author 2, C.D. Title of the article. Abbreviated Journal Name Year, Volume, page range.
2. Author 1, A.; Author 2, B. Title of the chapter. In Book Title, 2nd ed.; Editor 1, A., Editor 2, B., Eds.; Publisher: Publisher Location, Country, 2007; Volume 3, pp. 154–196.
3. Author 1, A.; Author 2, B. Book Title, 3rd ed.; Publisher: Publisher Location, Country, 2008; pp. 154–196.
4. Author 1, A.B.; Author 2, C. Title of Unpublished Work. Abbreviated Journal Name stage of publication
(under review; accepted; in press).
5. Author 1, A.B. (University, City, State, Country); Author 2, C. (Institute, City, State, Country). Personal communication, 2012.
6. Author 1, A.B.; Author 2, C.D.; Author 3, E.F. Title of Presentation. In Title of the Collected Work (if available), Proceedings of the Name of the Conference, Location of Conference, Country, Date of Conference; Editor 1, Editor 2, Eds. (if available); Publisher: City, Country, Year (if available); Abstract Number (optional), Pagination (optional).
7. Author 1, A.B. Title of Thesis. Level of Thesis, Degree-Granting University, Location of University, Date of Completion.
8. Title of Site. Available online: URL (accessed on Day Month Year).
